# Transcriptome and Gene Co-Expression Network Analysis Identifying Differentially Expressed Genes and Signal Pathways Involved in the Height Development of Banana (*Musa* spp.)

**DOI:** 10.3390/ijms24032628

**Published:** 2023-01-30

**Authors:** Bingyu Cai, Yixian Xie, Yufeng Chen, Miaomiao Cao, Junting Feng, Yuqi Li, Liu Yan, Yongzan Wei, Yankun Zhao, Jianghui Xie, Wei Wang

**Affiliations:** 1College of Tropical Crops, Hainan University, Haikou 570228, China; 2Key Laboratory of Biology and Genetic Resources of Tropical Crops, Ministry of Agriculture, Hainan Institute for Tropical Agricultural Resources, Institute of Tropical Bioscience and Biotechnology, Chinese Academy of Tropical Agricultural Sciences, Haikou, 571101, China; 3College of Horticulture, China Agricultural University, Beijing 100083, China

**Keywords:** banana, plant height, transcriptome, gene expression, gibberellic acid

## Abstract

Plant height is an important and valuable agronomic trait associated with yield and resistance to abiotic and biotic stresses. Dwarfism has positive effects on plant development and field management, especially for tall monocotyledon banana (*Musa* spp.). However, several key genes and their regulation mechanism of controlling plant height during banana development are unclear. In the present study, the popular cultivar ‘Brazilian banana’ (‘BX’) and its dwarf mutant (‘RK’) were selected to identify plant height-related genes by comparing the phenotypic and transcriptomic data. Banana seedlings with 3–4 leaves were planted in the greenhouse and field. We found that the third and fourth weeks are the key period of plant height development of the selected cultivars. A total of 4563 and 10507 differentially expressed genes (DEGs) were identified in the third and fourth weeks, respectively. Twenty modules were produced by the weighted gene co-expression network analysis (WGCNA). Eight modules were positively correlated with the plant height, and twelve other modules were negatively correlated. Combining with the analysis of DEGs and WGCNA, 13 genes in the signaling pathway of gibberellic acid (GA) and 7 genes in the signaling pathway of indole acetic acid (IAA) were identified. Hub genes related to plant height development were obtained in light of the significantly different expression levels (|log2FC| ≥ 1) at the critical stages. Moreover, GA3 treatment significantly induced the transcription expressions of the selected candidate genes, suggesting that GA signaling could play a key role in plant height development of banana. It provides an important gene resource for the regulation mechanism of banana plant development and assisted breeding of ideal plant architecture.

## 1. Introduction

Bananas (*Musa* spp.) are the most important staple food and fruit crops in tropical and subtropical regions [1,2]. They are the main export economic crop in developing countries, and their fresh fruit trade ranks first [3,4]. The main cultivated varieties of banana are almost all triploid, and only a few are diploids and tetraploids. The banana propagation is mainly dependent on asexual reproduction. Bananas have low lodging resistance due to their tall pseudostems, and are vulnerable to devastating damage by heavy wind and rain [5,6,7]. In addition, tall plants consume large amounts of water and nutrients during plant growth. Ideal plant height is conducive to mechanized planting, and decreased investment costs. Thus, dwarfing breeding is an important research direction for breeding excellent and high-yield varieties. Currently, the study of banana plant height is seriously restricted due to the complexity of the external morphology and ontogenesis of the pseudostem [8].

Numerous studies prove that dwarfism is an important plant architecture trait in crop breeding. It is undeniable that the dwarf plant has characteristics of lodging resistance, high planting density and convenient management [9]. It plays a positive role in management and harvesting strategies to increase production and labor efficiency [10,11]. Dwarf breeding is an important strategy in the crop breeding of rice, wheat, maize, etc. [12,13,14,15,16]. Since the 1960s, the planting of semi-dwarf varieties greatly increased crop yields, triggering the first green revolution [17,18,19]. Increasing evidence indicates that plant height is an important quantitative trait controlled by multiple genes [20,21]. Overexpression of the *TaERF8* gene in wheat can reduce plant height and increase yield [19]. The knock-down of *OsMPH1* and *OsFIE2* negatively regulates the plant height and yield [22]. *MiR529a*-targeting *SPL* (squamosa promoter binding-like) genes control plant height and grain size, further influencing production in rice [23]. Mutagenesis of *GmLHY* genes not only reduces plant height, but also shortens the internodes in soybean [24]. 

It is well known that hormones play a key role in regulating cell division, elongation and differentiation of the apical meristem and vascular bending tissue [18,25,26,27]. Several studies indicated that plant hormone biosynthesis and signaling transduction are highly correlated with plant growth and height regulation, including gibberellin (GA), indoleacetic acid (IAA), brassinosteroids (BRs) and cytokinin (CK) [28,29,30,31]. Mutations involved in IAA biosynthesis and signal transduction cause changes in plant height [32,33,34]. Repressing the expression of auxin-related genes effectively reduces plant height [35,36,37]. GAs take part in regulating many developmental processes such as cell elongation, seed germination, leaf expansion and fruit development [38,39,40,41]. Mutations in the biosynthetic and metabolic pathways of GAs exhibit the dwarfing phenotype [10]. Defects in terpene synthase (*TPS*) involved in the early step of GA biosynthesis cause plant dwarfism in maize and Arabidopsis [42,43,44]. Impaired *P450*-mediated GA biosynthesis has a negative impact on plant height development in maize and rice [45,46]. Additionally, IAA also induces gene expressions of GA biosynthesis, resulting in changes in plant height [47]. Excavation of functional genes can accelerate molecular breeding [48]. However, the limited gene and regulation mechanism of plant height still need to be further elucidated.

In this study, the critical period points of plant height differences in ‘BX’ and its mutant were determined. The ‘RK’ is a natural dwarf mutant of ‘BX’, with a stable phenotype in the field for many years. Based on the transcriptomic analysis and weighted gene co-expression network analysis (WGCNA), differentially expressed genes (DEGs) were identified between the selected five developmental stages. The important gene modules and hub genes related to plant height were predicted. Real-time PCR was performed to evaluate the expressions of important DEGs after treatment with GA3. This study provides a gene resource for further assaying the regulatory network of banana plant height.

## 2. Results

### 2.1. Analysis of Plant Height

Banana variety ‘RK’ is a natural mutation of ‘BX’. In the field, the plant height showed a stable significant difference at different years (Figure 1A,B). The tissue culture seedling of ‘BX’ and ‘RK’ were planted in the greenhouse, and plant height was measured once a week. In the greenhouse, no significant difference was observed in plant height between two varieties at the early stage of seedlings. The plant height of ‘BX’ was much higher than that of ‘RK’ after the fourth week, and the difference gradually increased with plant growth (Figure 1C–E). We speculated that the gene expression differences occurred after the third week. By contrast, the third and fourth weeks were the critical period for significant differences in plant height between two varieties.

### 2.2. Transcriptome Analysis of Banana Pseudostem Samples

The cDNA libraries of ‘BX’ and ‘RK’ pseudostem samples during different growth stages were prepared and sequenced using the Illumina 6000 platform. After filtering low-quality reads and removing adapter, a total of 207.22 Gb of clean data was obtained. The average clean data of each sample reached 4.23 Gb. The percentage of GC was among 47.59–53.76%. Q30 ranged from 93.21% to 95.88% (Appendix A), indicating that the quality of transcriptome data was relatively high. The clean reads were aligned with the banana reference genome. A total of 90.17–94.00% of the reads could be mapped, and 59.34–71.46% could be mapped to one location twice or more, which covered 32128 genes, and accounted for 91.08% of the whole genome (Appendix A).

### 2.3. Analysis of DEGs at Different Growth Stages

Compared to the transcriptome data of ‘RK’, there were 2593, 4999, 3294, 1997 and 1523 down-regulated genes, and 1970, 5508, 3223, 2205 and 2156 up-regulated genes from the third week to the seventh week in ‘BX’, respectively (Figure 2A and Table 1). Especially, 10507 of DEGs were detected in the fourth week, and then the numbers of DEGs decreased gradually. Comparing to the up-regulated DEGs of ‘BX’ and ‘RK’ in the third and fourth weeks, 1254 (‘BX’) and 1554 (‘RK’) DEGs were detected commonly, respectively Figure 2B,C).

### 2.4. Functional Enrichment Analysis of DEGs

The DEGs were annotated using GO and KEGG. A total of 1254 up-regulated DEGs in ‘BX’ and 1554 up-regulated DEGs in ‘RK’ between the third week and fourth week were significantly enriched (*p* < 0.05), respectively. There were 1656, 226 and 406 GO entries in ‘BX’, and 1831, 278 and 422 GO entries in ‘RK’ for the Biological Process, Molecular Function, and Cellular Component categories, respectively. The most highly enriched GO terms belonged to the GA-mediated signaling pathway (GO: 0009937, GO: 0009939, GO: 0009740, 13 DEGs) in ‘BX’ (Figure 3A). The most highly enriched GO terms were mitotic cell cycle (GO: 1903047, 76 DEGs), organelle fission (GO: 0048285, 71 DEGs), regulation of cell cycle (GO: 0051726, 60 DEGs) and other related GO terms in ‘RK’ (Figure 3B). The most highly enriched KEGG terms were MAPK signaling pathway (ko04016, 31 DEGs) and Indole alkaloid biosynthesis (ko00901, 7 DEGs) in ‘BX’ (Figure 2C), while the most highly enriched KEGG terms were cell cycle (ko04110, 39 DEGs) and DNA replication (ko03030, 20 DEGs) in ‘RK’ (Figure 3D). The differential expressions of GA and IAA-related genes may be one of the main reasons for causing the difference in plant height between ‘BX’ and ‘RK’.

### 2.5. Analysis of GA Signaling Pathway

To further analyze the function of DEGs, KEGG was used to locate the DEGs in plant hormone signal transduction pathways. The results showed that nine of the thirteen genes from regulation of the GA-mediated signaling pathway term were the homologous gene of gibberellin receptors GA-INSENSITIVE DWARF1 (GID1), and the other four genes were not annotated (Figure 4B). In the presence of GA signal, GID1 indirectly controlled stem growth and induced germination by regulating DELLA and its downstream PIF4/bHLH [49] (Figure 4A). Six DELLA genes were identified by the eggNOG annotation. Expression data analysis showed that DELLA of ‘BX’ and ‘RK’ had different expression levels after the third week, and exhibited a high transcription accumulation in the whole growth period (Figure 4C). Especially for DELLA4, the expression levels showed the most significant differences between ‘BX’ and ‘RK’ at the fifth week.

### 2.6. Construction of Gene Co-Expression Network

To further reveal transcriptome changes in different growth stages, WGCNA was used to analyze the transcription levels of ‘BX’ and ‘RK’ at different growth stages. After filtering, a total of 17922 genes were selected for WGCNA analysis. A total of 20 modules with different colors were identified, and grey represented no assignment to any module. Significant differences were found between different modules (Figure 5A,B). Expression patterns were correlated with physiological trait data, and different modules were analyzed using the correlation analysis with the plant height. Eight modules were positively correlated with plant height, and twelve of them were negatively correlated with plant height. The white module (r = 0.69) and midnightblue module (r = −0.83) were the most positively and negatively correlated with plant height, respectively (Figure 5C).

The modules of bisque4 and salmon were positively related to plant height, and contained 1549 and 1560 genes, respectively. Combining with DEG analysis, six genes from GA-mediated signaling pathway (GO: 0009937, GO: 0009939, GO: 0009740) and two genes from IAA biosynthesis (ko00901) were obtained from the bisque2 module (Figure 6A). Two genes involved in the GA signaling pathway were identified from the salmon module. Based on the eggNOG function annotation, some GA- and ethylene-related genes were significantly enriched in both the bisque4 and salmon modules (Figure 6B and Appendix A). A total of 52 and 35 hub genes were identified from the bisque4 and the salmon, respectively. One GA receptor gene was identified from the hub genes of bisque4 module. DEGs in ‘BX’ and ‘RK’ were significantly different from the third week to the seventh week, especially in the fourth week (Figure 6D).

As the most positively correlated module with plant height, a total of 116 and 193 high correlation genes were identified in the white and midnightblue modules, respectively. In the white module, six candidates, including three *MYB* transcription factors, one *HLH*, one *GID1C* and one *ARF*, were identified. The *GID1C* was also found from up-regulated DEGs in ‘BX’ between the third week and the fourth week. The most highly enriched GO terms were cell cycle (GO: 0051726, 7 genes) and intracellular steroid hormone receptor signaling pathway (GO: 0030518, 2 genes). Cytoscape was used to analyze the network in the white module, and a total of 17 hub genes were identified. These genes showed significant differences at different growth stages from the third week to the fifth week in ‘BX’ and ‘RK’ (Appendix A). *GID1*, *ARF* and 21 hub genes were considered as candidate central genes for the white module.

The midnightblue module was the most negatively correlated with plant height; we identified 17 ATP synthases, 12 NADH, 4 *MYB*, 4 stress transcription factor and 2 mitochondrial ATP synthases. The most highly enriched GO terms contained mitochondrial membrane (GO:0031966, 31 genes), respiratory chain complex (GO:0098803, 21 genes), NADH dehydrogenase complex (GO:0030964, 15 genes), ribose phosphate metabolic process (GO:0019693, 15 genes) and ATP metabolic process (GO:0046034, 10 genes). A total of 21 hub genes were obtained, including 3 ATP synthases (Appendix A).

### 2.7. Determination of Candidate Gene Expressions Using qRT-PCR

To validate the RNA-seq analysis results, nine genes of GA-mediated signaling pathway were selected for qRT-PCR analysis. The qRT-PCR results showed that the selected nine genes were significantly up-regulated (*p* < 0.01) in the pseudostem of ‘BX’ treated with GA3 after one hour (Figure 7). In the fourth week, seven genes showed much higher transcription levels in ‘BX’ than those in ‘RK’, except for *MaGAR1* and *MaGAR8* (Appendix A). These results were consistent with the transcriptome analysis.

## 3. Discussion

In the early research, new dwarf and semi-dwarf varieties were obtained mainly through hybridization of tall and dwarf varieties in crops such as wheat, maize and rice [50,51,52]. Hybridization is an effective breeding method to improve specific traits, but it takes a long time. With the development of molecular breeding, the study of functional genes speeds up the breeding process, and shortens the breeding time [48,53,54]. Identifying the molecular genetic basis of plant height is conducive to breeding new varieties with ideal plant height, via the molecular breeding strategy. The ideal plant height can improve the lodging resistance of bananas, adapt to mechanization and increase production potential. Banana cultivars are almost all triploid, and rapid propagation is a main method using tissue culture. The new dwarf banana cultivars were obtained mainly through natural variation [55]. Additionally, the dwarf varieties of ‘8188-1’ and ‘Zhongjiao NO.12’ were obtained from ‘Brazilian banana (*Musa* spp. AAA group)’ by chemical and physical mutagenesis, respectively [16]. A new dwarf banana cultivar (‘Aifen No.1’) came from ‘Pisang’, by observing phenotypes in tissue culture. The plant height of ‘Aifen No.1’ was about 2.0–2.5 m significantly lower than that of wild type (4.5–4.8 m) [16]. However, bud mutation and mutagenesis cost a lot of time, and are uncontrollable, which greatly limits the process of banana breeding. Thus, excavating plant height-related genes and analyzing plant height formation mechanism is an effective method to solve dwarf breeding challenges of banana. The dwarf mutants of banana are important materials for identifying and searching dwarf-related genes. In this study, the popular cultivar ‘Brazilian banana’ (‘BX’) and its dwarf mutant (‘RK’) were selected to identify the plant height-related genes. In the greenhouse, no significant difference was observed in plant height between two varieties at the early stage of seedlings. Plant height showed a very significant difference in the fourth week, and the difference gradually increased with plant growth. Therefore, the third and fourth weeks were the critical periods for significant differences in plant height between two varieties, and DEGs controlling plant height occurred after the third week.

Plant height is a quantitative trait that is controlled by multiple genes [56]. In this study, the DEGs of GA biosynthesis were not recorded in the comparative transcriptome analysis between ‘BX’ and ‘RK’ banana pseudostems. After GO enrichment analysis, a total of 13 genes were found to be involved in the signaling pathway of GA, of which 9 genes were homologous genes of *GID1*. The functions of the other four genes were unknown. It suggested that the difference in plant height between ‘BX’ and ‘RK’ may be caused by different gene expressions of GA signal transduction. Previous studies also showed that mutations involved in GA biosynthesis and metabolic pathways led to dwarf phenotypes in plants [10]. Thus far, more than 130 GAs have been found, and only a few of them have biological activity, including GA1, GA3, GA4 and GA7 [57]. In the GA signal transduction pathway, GIBBERELLIN INSENSITIVE DWARF1 (*GID1*) and *DELLA* proteins play an important role [58]. Bioactive GAs are first received by *GID1*, binding and activating downstream *DELLA* proteins [59]. Then, the *DELLA* binds to the *E3* ubiquitin ligase SLEEPY1 (*SLY1*) and triggers its degradation by the 26S proteasome pathway [60]. After degradation, the downstream genes of signal transduction are activated to regulate the plant developmental process, including squamosa promoter-binding protein-likes (*SPLs*), GAMYB, SUPPRESSOR OF CONSTANS 1 (*SOC1*), and phytochrome-interacting factors (*PIFs*), and the downregulation of SPINDLY (*SPY*) [56,61,62]. Similarly, combining phenotype and transcriptome, a total of eight GA signal transduction DEGs were detected in the modules that were positively related with plant height by WGCNA, in which the homologous gene of *GID1* was identified as the hub gene from the positively module. In addition, downstream *DELLA* genes showed significant differences between ‘BX’ and ‘RK’. All of the *GID1* homologous genes were significantly up-regulated after GA3 treatment, suggesting these genes were involved in GA signal transduction. Our results further supported that GA signal transduction is critical for plant height.

In addition to hormone signal transduction pathways, GA also interacts with other hormones to regulate plant development, including IAA, BR, CK and ABA. The bioactive auxin and response factors positively regulated GA signaling and biosynthesis, but negatively regulated *DELLA*, including AUXIN RESPONSE FACTOR (*ARF*) family and AUXIN INDUCIBLE/INDOLE-3-ACETIC ACID INDUCIBLE (*AUX/IAA*) [37,63,64]. In our present study, seven genes of the IAA signal pathway were identified from the modules that were positively associated with plant height through WGCNA. This indicated that IAA biosynthesis was also involved in the regulation of plant height development. In addition, one *ARF* was found in the white module positively associated with plant height, and *GID1* was also involved in this module (Appendix A). Therefore, auxin response factors were important for the positive regulation of plant height. These findings suggested that plant growth was regulated by IAA and GA. In addition, we also found several genes in the MAPK signaling pathway and cell cycle that also play important roles in the formation of plant height [64]. Our research provides a foundation for further study on the mechanism of banana dwarfing.

## 4. Materials and Methods

### 4.1. Plant Material

The banana cultivar Brazilian (*Musa* spp. AAA group, ‘BX’) and its mutant (Reke 3) were selected as materials. The seedlings from vegetative propagation were planted in the experimental field and greenhouse of the Chinese Academy of Tropical Agricultural sciences in Haikou, Hainan, China. The plant height of seedlings was measured and recorded once a week from planting.

### 4.2. Collection of Banana Pseudostem Samples for RNA-Sequencing

The banana pseudostem tissue was collected weekly from the plant middle of ‘BX’ and ‘RK’. Three plants were used at each time point. The selected samples were immediately stored at −80 °C. 

### 4.3. Collection of Banana Pseudostem Samples for RNA-Sequencing

Total RNA was extracted from banana pseudostem using the RNA extraction kit (TIANGEN, Beijing, China). Genomic DNA contamination was removed with RNase-free DNase I. RNA degradation and contamination were assessed using 1.2% agarose gel electrophoresis. The integrity and concentration of RNA were detected using a NanoDrop One (Thermo, Wilmington, America). A total amount of 3 μg RNA was used to construct each cDNA library. A total of 48 RNA libraries from eight growth stages were constructed. All libraries were sequenced on the Illumina 6000 platform by the BioMarker Technologies Company (Beijing, China).

### 4.4. Transcriptome Analysis

The Q20, Q30 and GC content were calculated using the FastQC (v0.11.9) software. The adapter, ploy-N and low-quality reads were filtered according to the quality control by the fastp (v0.20.0) software. The clean reads were aligned with the banana reference genome v2 download from the Banana Genome Hub (https://banana-genome-hub.southgreen.fr/ accessed on 6 December 2022). Hisat2 (v2.1.0) and QualiMap (v.2.2.1) were used to analyze the genome coverage and mapping rate. The gene expression levels were analyzed by the htseq-count (v1.99.2), and then converted the gene count to the FPKM (expected number of fragments per kilobase of transcript sequence per million base pairs sequenced). DEGs between samples of ‘BX’ and ‘RK’ at different growth stages were identified using the DESeq2 R package (v1.32.0) according to the standard of FC ≥ 2, FDR < 0.01 and *p* < 0.05.

GO enrichment of DEGs was analyzed using the eggNOG-mapper (v1.0.3) against the reference database (http://eggnog5.embl.de/download/emapperdb-5.0.1/; accessed on 20 March 2022). The protein function was annotated, and the GO and KEGG of DEGs were analyzed using the clusterfiler in R package.

### 4.5. Identification of WGCNA Modules Related with Plant Height

The co-expression network was analyzed using WGCNA in R package. Growth period datasets of ‘BX’ and ‘RK’ were filtered to remove the average value of FPKM less than 5. The filtered FPKM values were used to construct co-expression networks. Genes were grouped according to the topological overlap dissimilarity measure (1-TOM) of gene connection strength by average hierarchical clustering. The network modules used the dynamic tree cut algorithm with the minimum cluster size of 30 and the merging threshold function of 0.80 for identification. To link the physiological measurement with the network, the module eigengenes were associated with the physiological data. The GO and KEGG enrichment of each module was analyzed using clusterfiler in R package. The hub genes of modules highly related to phenotype were identified using cytoscape (v3.8.2). The hub genes were identified through 12 algorithms of cytoHubba (v0.1).

### 4.6. Exogenous Hormone Treatment

To further validate the reliability of candidate genes, GA3 was used for spraying the whole banana plant in the greenhouse of the Chinese Academy of Tropical Agricultural sciences. The concentration of GA3 was set as 20 mg/L, with sterile water as the control. The whole plant of banana seedlings was sprayed two weeks after planting. The banana pseudostem tissue was collected at 0 h, 1 h, 6 h, 12 h and 24 h after treatment, respectively. At least three plants from each group were collected randomly and were stored at −80 °C.

### 4.7. qRT-PCR Analysis

For the quantitative RT-PCR analysis, total RNA was extracted using the RNA extraction kit, and a reverse transcription kit (Takara, Kusatsu, Japan) was used for the first complementary DNA (cDNA) synthesis. The RT-PCR primers of candidate genes (Appendix A) were designed using primer blast (https://www.ncbi.nlm.nih.gov/tools/primer-blast; accessed on 17 May 2022). Diluted cDNA was amplified using TB Green premix EX Taq II. PCR was performed on a LightCycler96 Real-Time System (Roche, Basel, CH). Each sample was performed with three independent biological replicates. The actin gene was used as an internal control [21].

### 4.8. Statistical Analysis

For statistical analysis, at least three samples were taken to calculate the average ±SD. Statistical analysis was performed using Student’s *t*-test within SPSS 22.0 (Chicago, IL, United States). Statistically significant differences were indicated with *p* < 0.05 (*).

## 5. Conclusions

In this study, the key period of plant height development of ‘BX’ and its dwarf mutant ‘RK’ were identified via phenotypic analysis. A total of 13 candidate genes in the signaling pathway of GA, and 7 candidate genes in the signaling pathway of IAA were found with WGCNA and transcriptome analysis in the critical period. Especially, the GA-related genes were induced significantly after GA3 treatment. Those genes showed higher expression levels in ‘BX’ than ‘RK’ at the fourth week, and were consistent with the transcriptome analysis.

## Figures and Tables

**Figure 1 ijms-24-02628-f001:**
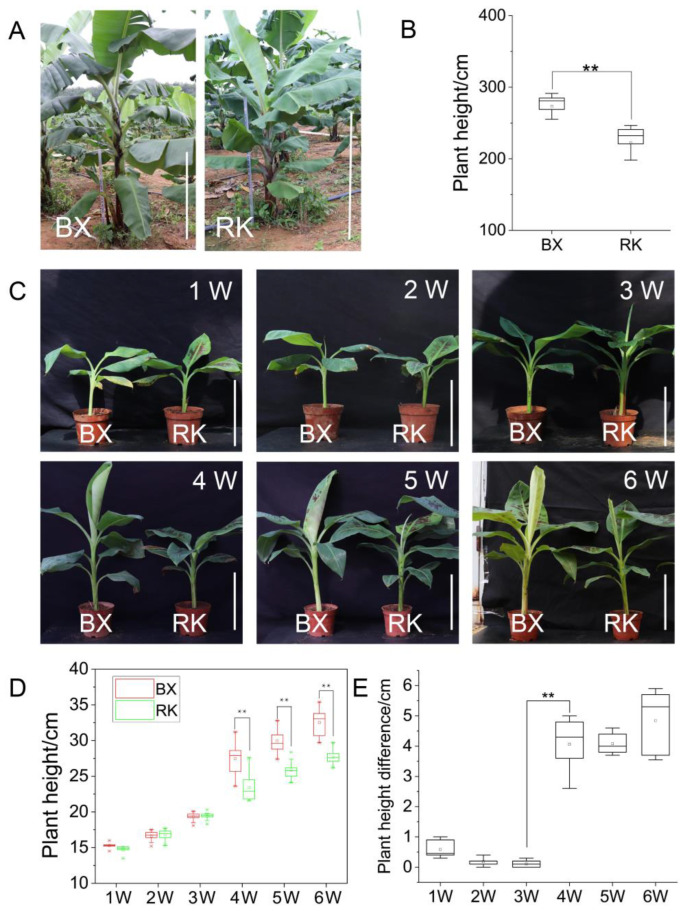
Changes in plant height during the growth between ‘BX’ and ‘RK’. (**A**) Phenotypes of ‘BX’ and ‘RK’ in the field. Bars = 1 m; (**B**) box plot of plant height in field of ‘BX’ and ‘RK’ (**, *p* ≤ 0.01); (**C**) growth changes of ‘BX’ and ‘RK’ from the first week to the sixth week in the greenhouse. Bars = 20 cm; (**D**) measurement of plant height between ‘BX’ and ‘RK’ from the first week to the sixth week in the greenhouse (**, *p* ≤ 0.01); (**E**) differences in plant height between ‘BX’ and ‘RK’ during the six weeks after planting.

**Figure 2 ijms-24-02628-f002:**
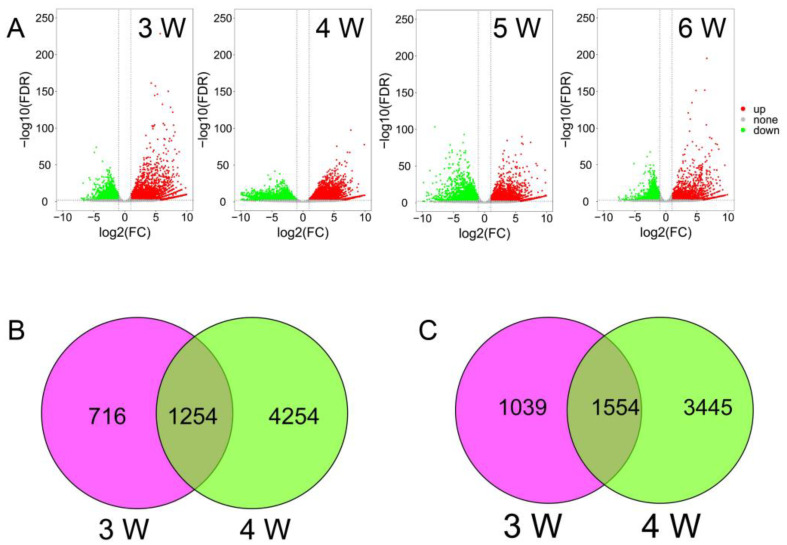
DEG analysis between ‘BX’ and ‘RK’ at different growth stages. (**A**) DEGs (FDR, *p* ≤ 0.01, FC ≥ 2) from the third week to the sixth week. Red and green points represent the up-regulated and down-regulated genes in ‘BX’, respectively; (**B**) numbers of up-regulated genes between the third week and fourth week in ‘BX’; (**C**) numbers of up-regulated genes between the third week and fourth week in ‘RK’.

**Figure 3 ijms-24-02628-f003:**
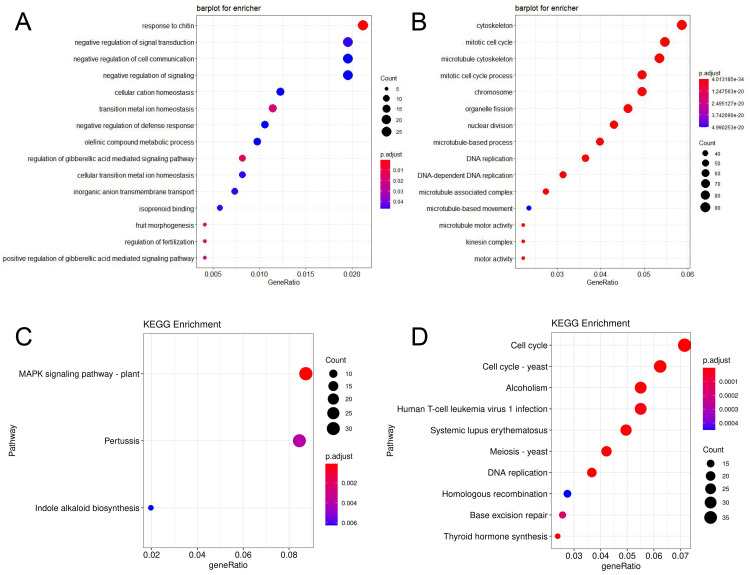
Enriched GO and KEGG of up-regulated DEGs in ‘BX’ and ‘RK’. (**A**) GO enrichment terms of up-regulated DEGs in ‘BX’; (**B**) GO enrichment terms of up-regulated DEGs in ‘RK’; (**C**) KEGG enrichment of up-regulated DEGs in ‘BX’; (**D**) KEGG enrichment of up-regulated DEGs in ‘RK’. The sizes of points represent the numbers of genes.

**Figure 4 ijms-24-02628-f004:**
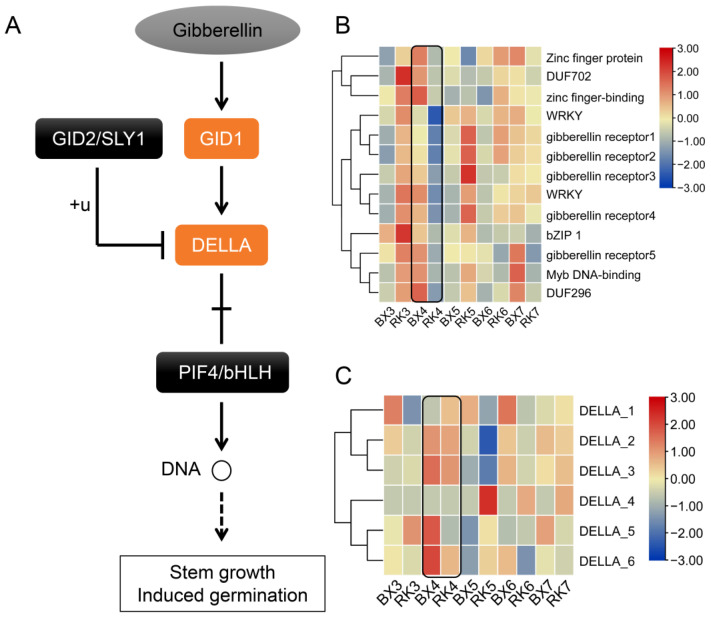
The expressions of DEGs in GA signal transduction pathway at different growth stages. (**A**) GA signal transduction pathway; (**B**) expression heatmap of DEG expressions in gibberellin receptor GID1; (**C**) expression heatmap of *DELLA* genes. The color scale ranging from blue to red represents the expression levels of genes.

**Figure 5 ijms-24-02628-f005:**
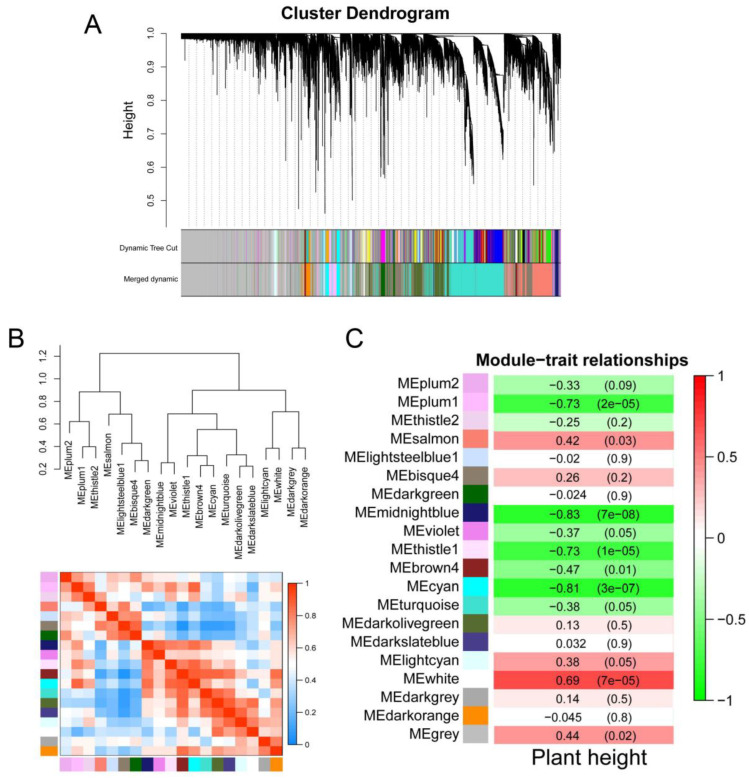
WGCNA at different growth stages of ‘BX’ and ‘RK’. (**A**) Hierarchical clustering tree (cluster dendrogram) of all expressed genes. Co-expression modules were identified by WGCNA at different growth stages. Each leaf in the tree represents one gene. The branches correspond to modules of highly interconnected groups of genes. The different colors represent different gene modules. The major tree branches constituted 20 modules; (**B**) module-relatedness clustering and heatmap of correlation in each module; (**C**) heatmap chart showing module-plant height correlation and *p*-values. The color scale ranging from green to red indicates negative and positive correlations between the module and plant height. The numbers in brackets represent the *p*-values.

**Figure 6 ijms-24-02628-f006:**
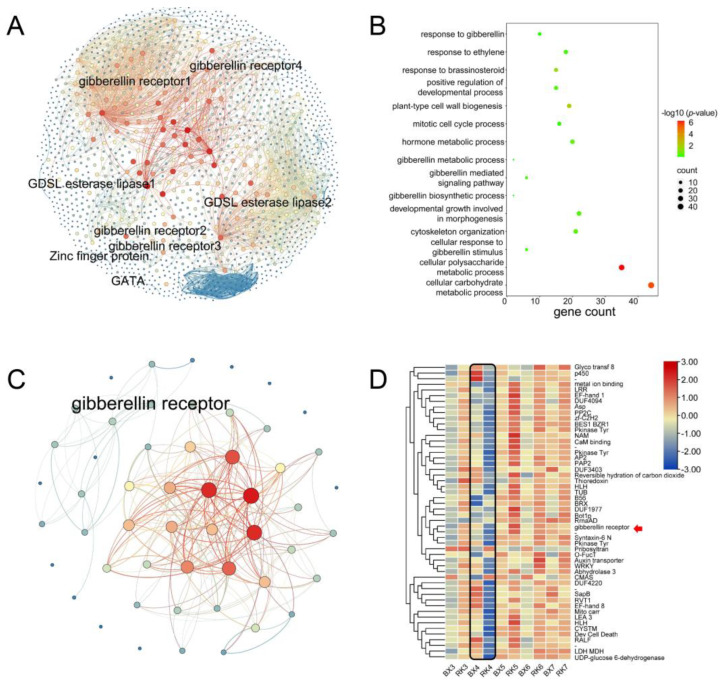
Co-expression network analysis of genes in the bisque4 module. (**A**) Co-expression network of genes in the bisque4 module. The size of nodes represents the number of genes. The width of edges represents the weight between two genes connected; (**B**) GO enrichment analysis of the bisque4 module. The size of the point represents the number of genes. The colors ranging from green to red represent the *p*-values; (**C**) merging network of hub genes using 12 methods; (**D**) expressions of hub genes from the third week to the seventh week. The color scale ranging from blue to red represents the expression levels of genes. The red arrow indicates the gibberellic receptor.

**Figure 7 ijms-24-02628-f007:**
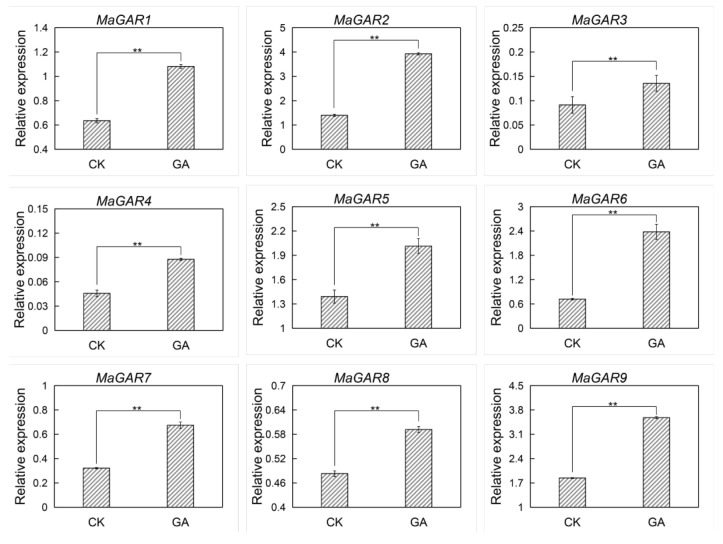
Expression levels of candidate genes after GA3 treatment using qRT-PCR. Data were means of three biological repetitions. Each bar represents mean ± SE, and asterisks (**) indicate significant differences (Student’s *t*-test, *p* < 0.01).

**Table 1 ijms-24-02628-t001:** DEGs at different growth stages of ‘BX’ and ‘RK’.

Sample	Down-Regulation	Up-Regulation	Total DEGs
BX3_vs_RK3	2593	1970	4563
BX4_vs_RK4	4999	5508	10507
BX5_vs_RK5	3294	3223	6517
BX6_vs_RK6	1997	2205	4202
BX7_vs_RK7	1523	2156	3679

## Data Availability

The original contributions presented in the study are publicly available. This RAN-seq raw data can be found in the NCBI repository, accession number: PRJNA910085.

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
