# Peer review of "Transcriptome and Gene Co-Expression Network Analysis Identifying Differentially Expressed Genes and Signal Pathways Involved in the Height Development of Banana (Musa spp.)"

_ijms, 2023, doi:10.3390/ijms24032628_

Round 1

Reviewer 1 Report

In this manuscript, authors compared two banana cultivars at the phenotypic and transcriptomic levels, which revealed some candidate genes responsible banana height and their co-expression network. This work will provide fundamental support for banana breeding. Overall, the manuscript was presented in high quality while there are some minor comments for authors. 

1. Could you please provide some genetic background of RK in the introduction, if available?

2. In figure 7, the authors validated some candidate genes under GA treatment while they didn't validate the expression of the candidate genes under the same treatment as the sample of transcriptomics analysis. The latter is more important to validate the differential expression. 

Author Response

Dear editor,  

First of all, I would like to thank you and reviewers to take time to deal with our manuscript (ijms-2117215). We highly appreciate the favorable assessment for our work and are happy to submit a revised version of our manuscript. According to your and reviewers’ suggestions, we provide a point-by-point explanation of the changes made and our response to your and reviewers' comments.

Reviewer #1

Point 1 of Reviewer #1: Could you please provide some genetic background of RK in the introduction, if available?

Response to the comment: Thanks for your suggestions, the authors added the genetic background of RK in the Introduction (Line 82-83).

Point 2 of Reviewer #1: In figure 7, the authors validated some candidate genes under GA treatment while they didn't validate the expression of the candidate genes under the same treatment as the sample of transcriptomics analysis. The latter is more important to validate the differential expression.

Response to the comment: Considering the Reviewer’s suggestion, the author added the expression analysis of these candidate genes. Based on the growth of plants in different weeks, we found that the fourth week is the key time point of plant height development in the selected cultivars. Therefore, the expression characteristics of these candidate genes were analysis using qRT-PCR in the fourth week. The results were added in the Supplementary Data (Figure S4).

As you see from our revision in the revised version, we addressed the issues raised by editor and reviewers in the text. These changes were marked with red and did not influence the content and framework of the paper. We appreciate for your and Reviewers’ warm work earnestly, and hope that the correction will meet with approval.

Once again, thank you very much for your comments and suggestions.

With kind regards,

Dr. Wei Wang

Reviewer 2 Report

In this study, using the transcriptomic data and differentially expressed genes (DEGs) the authors have identified key developmental stages. Moreover, several important gene modules and hub genes related to plant height are predicted. The qRTPCR analysis indicated the expression of DEGs after treatment with GA3. Overall, the study provides a plant height specific genetic resource for network analysis in banana plants.

I have few comments:

-Some redundant usage of words could be avoided. For instance, important word is noticeably overused.

The discussion two paras, line 244 to 270 and line 271-296, needs to be reorganized.

Make shorter paragraph and include recent studies for discussion and also introduction:

Assessment of genetic diversity and volatile content of commercially grown banana (Musa spp.) cultivars, Hinge et al., Scientific Reports, 2022; https://doi.org/10.1038/s41598-022-11992-1 (Banana); Microsatellite and RAPD analysis of grape (Vitis spp.) accessions and identification of duplicates/misnomers in germplasm collection, Upadhyay et al., 2010 Indian J Hortic Volume 67 Pages 8-15; Microsatellite analysis to differentiate clones of Thompson seedless grapevine, Upadhyay et al., 2010, Ind Journal of Horticulture, Volume 67 Issue 2 Pages 260-263.

 A conclusions section is missing.

 Overall, the study looks promising and suggested correction will improve the manuscript further.

Author Response

Dear editor,  

First of all, I would like to thank you and reviewers to take time to deal with our manuscript (ijms-2117215). We highly appreciate the favorable assessment for our work and are happy to submit a revised version of our manuscript. According to your and reviewers’ suggestions, we provide a point-by-point explanation of the changes made and our response to your and reviewers' comments.

Reviewer #2

Point 1 of Reviewer #1: Some redundant usage of words could be avoided. For instance, important word is noticeably overused.

Response to the comment: Thanks for the Reviewer’s suggestions, the authors deleted some redundant words in this article and simplified several sentences.

Point 2 of Reviewer #1: The discussion two paras, line 244 to 270 and line 271-296, needs to be reorganized.

Response to the comment: Considering the Reviewer’s suggestion, the authors reorganized the discussion in lines 244 to 270 and lines 271-296.

Point 3 of Reviewer #1:  Make shorter paragraph and include recent studies for discussion and also introduction: Assessment of genetic diversity and volatile content of commercially grown banana (Musa spp.) cultivars, Hinge et al., Scientific Reports, 2022; https://doi.org/10.1038/s41598-022-11992-1 (Banana); Microsatellite and RAPD analysis of grape (Vitis spp.) accessions and identification of duplicates/misnomers in germplasm collection, Upadhyay et al., 2010 Indian J Hortic Volume 67 Pages 8-15; Microsatellite analysis to differentiate clones of Thompson seedless grapevine, Upadhyay et al., 2010, Ind Journal of Horticulture, Volume 67 Issue 2 Pages 260-263.

Response to the comment: Considering the Reviewer’s suggestion, the authors added some recent references in the Introduction and discussion.

Point 4 of Reviewer #1: A conclusions section is missing.

Response to the comment: According to your suggestions, the authors added a conclusion in lines 386-391.

As you see from our revision in the revised version, we addressed the issues raised by editor and reviewers in the text. These changes were marked with red and did not influence the content and framework of the paper. We appreciate for your and Reviewers’ warm work earnestly, and hope that the correction will meet with approval.

Once again, thank you very much for your comments and suggestions.

With kind regards,

Dr. Wei Wang

Round 2

Reviewer 2 Report

Authors have revised the manuscript satisfactorily.